# “Buy High, Sell Low”: A Qualitative Study of Cryptocurrency Traders Who Experience Harm

**DOI:** 10.3390/ijerph20105833

**Published:** 2023-05-16

**Authors:** Benjamin Johnson, Tianze Sun, Daniel Stjepanović, Giang Vu, Gary C. K. Chan

**Affiliations:** 1National Centre for Youth Substance Use Research, The University of Queensland, Brisbane, QLD 4072, Australia; 2School of Psychology, The University of Queensland, Brisbane, QLD 4072, Australia

**Keywords:** mental health, cryptocurrency, financial trading, gambling, addiction

## Abstract

The constant, substantial price fluctuations of cryptocurrency allow traders to engage in highly speculative trading that closely resembles gambling. With significant financial loss associated with adverse mental health outcomes, it is important to investigate the impact that market participation has on mental health. Therefore, we conducted interviews with 17 participants who self-reported problems due to trading. Thematic analysis was conducted revealing themes: (1) factors in engagement, (2) impacts of trading and (3) harm reduction. Factors in engagement captured factors that motivated and sustained cryptocurrency trading. Impacts of trading outlined how cryptocurrency trading positively and negatively impacted participants. Harm reduction described methods participants employed to reduce mental distress from trading. Our study provides novel insights into the adverse impacts of cryptocurrency trading across multiple domains, especially mental health, relationships and finances. They also indicate the importance of further research on effective coping strategies for distress caused by financial loss from trading. Additionally, our study reveals the significant role social environments play on participants’ expectations and intentions regarding cryptocurrency trading. These social networks extend beyond real-life relationship to include celebrity and influencer endorsement. This encourages investigation into the content of cryptocurrency promotions and the influence they have on individuals’ decision to trade.

## 1. Introduction

### 1.1. The Cryptocurrency Market

Cryptocurrencies are digital software designed to function as electronic money. Whilst they have not displaced fiat currencies, they have instead gained popularity as high-risk investments known for their tremendous growth and volatility. The cryptocurrency market’s most substantial rise came in 2021, which saw its market cap grow from USD 400 billion in November 2020 to USD 2.8 trillion a year later [1].

The rapid growth in cryptocurrency has widely gained the attention of investors. Individual investors can gain exposure to cryptocurrency through multiple avenues, such as exchange-traded funds (ETFs), direct buying from a seller and, most commonly, through cryptocurrency exchanges such as Binance and Coinbase. These exchanges have been enormously popular, with Coinbase reporting revenues of USD 7.4 billion in 2021, larger than the New York Stock Exchange (7.1 billion) and the Nasdaq (5.8 billion) [2]. These trading applications allow users to buy and sell various cryptocurrencies, with Binance supporting trading for over 600 coins [3].

Through these exchanges, many traders engage in speculative trading patterns that heavily resemble gambling [4]. These may be especially attractive for problem gamblers, with findings suggesting they may engage in cryptocurrency trading as an alternative gambling activity [5]. This includes day trading, a high-risk strategy that involves the rapid buying and selling of cryptocurrencies to capitalize on short-term price fluctuations. Also popular is leverage trading, where traders increase their position size by borrowing third-party funds against their initial capital. Whilst this amplifies potential returns, it also increases the potential for loss. Furthermore, if the price of an asset moves significantly against the leveraged position, traders are at risk of forced liquidation, resulting in the loss of their deposited funds.

This has led to some investigation into the relationship between cryptocurrency trading and gambling [6]. Studies investigating problem gamblers revealed associations between problem gambling symptoms and cryptocurrency trading intensity [7]. These studies also indicate problem gamblers may engage in cryptocurrency trading as another form of gambling activity, as suggested by correlations between cryptocurrency trading and other gambling activities [8]. However, the extent to which trading leads to addiction, similar to non-substance addiction in gambling, remains speculative due to limited empirical research. Further research is needed on the experiences of individuals with problematic cryptocurrency trading to reveal the drivers, symptoms, and consequences of their trading.

Additionally, research should look to examine cryptocurrency investors in general, with the risk of economic loss not exclusive to speculative traders. Without proper regulation, investors are vulnerable to financial crime and market manipulation, which can result in significant financial losses. Additionally, the cryptocurrency market is still marred with volatility, regularly experiencing 50% drawdowns in overall market capitalization [1]. The most recent fall has been no exception, with the market plunging off its highs in November 2021 by over 70% as of December 2022 [1].

### 1.2. Mental Health and Existing Research

Hazardous and turbulent, the cryptocurrency market may expose traders to a high risk of economic loss. Large economic loss is associated with adverse mental health outcomes such as depression and anxiety [9,10]. Additionally, volatile markets have been found to significantly increase the anxiety level of investors, regardless of the general trend of prices [11]. Given its popularity, especially amongst young people [12,13], cryptocurrency trading may be a concern for public health and warrants investigation.

Preliminary findings partially validate these concerns. A population survey of Finish residents found that cryptocurrency traders reported higher psychological distress and loneliness than monthly stock investors and non-investors [8]. Additionally, Mills and colleague’s cohort study of frequent gamblers found associations between cryptocurrency trading frequency with anxiety and depression symptoms [7].

However, anxiety and depression scores were found to be non-significant predictors when controling for participation in gambling activities such as sports betting or slots [7]. In addition, these findings are limited in their comprehensiveness since they were based on scores from abbreviated measures. Furthermore, these findings were contradicted by Kim and colleagues’ study, which found that Bitcoin investors scored in the low-risk category for trait anxiety and mood disorder and were not significantly different from share investors and non-investors [13].

The cross-sectional designs of these studies mean we cannot assess how cryptocurrency trading impacts individuals over a more extended period. Additionally, these studies do not consider the current performance of the cryptocurrency market, which may influence scores on mental health measures. This may explain the findings of Kim and colleagues’ study [13], which surveyed investors during a week when Bitcoin rose by 18% [1].

### 1.3. The Study

Our current understanding of the psychological impact of cryptocurrency trading is insufficient, given its potential impact on mental health. With adverse health outcomes heavily pronounced following stock market crashes [14], the recent cryptocurrency downturn provides a timely opportunity to investigate its effects on market participants.

Therefore, in this study, we conducted a semi-structured qualitative interview of cryptocurrency traders who have experienced harm due to their trading. This has not been previously done, with existing research exclusively quantitative, leaving the personal experiences of cryptocurrency traders unexplored.

Qualitative studies of risky financial behaviours, such as gambling, have provided important insights into their impact on individuals across various domains [15,16]. By employing a qualitative approach, our study aimed to do likewise.

Our main objective was to examine the impact that cryptocurrency trading has on mental health and psychological wellbeing. We also aimed to capture its effects across other domains of life, such as relational and occupational. Our secondary objectives were to capture the factors influencing individuals’ exposure and severity of harm and to investigate how individuals cope with personal distress incurred by trading.

## 2. Materials and Methods

### 2.1. Subject Recruitment

Participants were recruited via online advertisement between April to June 2022. We advertised on two platforms: the largest English cryptocurrency discussion forum, r/cryptocurrency, and the University of Queensland website.

Our inclusion criteria were adult individuals who had experienced problems due to their cryptocurrency trading. We wanted to recruit individuals with self-reported cryptocurrency trading problems to ensure we adequately captured the experiences of individuals who were or had experienced harm or negative outcomes from their cryptocurrency trading behaviour. Individuals could express interest through various contact methods (e.g., Email, Telegram, Twitter), allowing them to remain anonymous. Eligible participants were invited to contact us to participate in an online interview. They were assigned randomized numbers (e.g., 25K) to help obscure their identities and keep their answers distinct in analysis. We also distributed an information and consent sheet, which provided further information about the study and assurances of confidentiality and anonymity.

### 2.2. Data Collection

Interviews were carried out between June and July 2022, and were conducted and recorded via Zoom conferencing. The median duration of interviews was 42 min, ranging between 20 and 63 min. We continued interviews until we achieved data saturation.

The interview process was guided by prepared questions that can be found in Appendix A. However, it followed a semi-structured design, meaning we could stray from our set questions based on the information that the interviewees brought up. This allowed us to deeply explore topics of interest to the interviewer and interviewee, and uncover information in the natural flow of conversation.

We began by gathering verbal consent and asking standardized demographic questions. We then followed by asking participants a series of open-ended questions about their cryptocurrency trading experiences. These were regarding participants cryptocurrency trading behaviours and the impacts of their trading. We also asked participants about their motivations for cryptocurrency trading.

At the end of the interview, we thanked participants and asked if they had any additional comments. Participants were reimbursed for their time through a gift card equivalent to 20 AUD.

### 2.3. Sample Characteristics

We interviewed 20 participants in total; however, data from two interviews were unsuitable due to connectivity issues. Additionally, one participant was interviewed as a pilot to test the suitability of the questions. This left our final sample at 17, within the typical sample size in qualitative interview studies [17]. Our sample included 13 males and four females, ranging from 20 to 32 years old, with a median age of 25. All participants resided in the United States (*n* = 12) or Australia (*n* = 5). Full demographic characteristics can be seen in Table 1.

### 2.4. Data Analysis

Audio files were automatically transcribed through the program Otter.ai. We also manually checked transcripts to ensure their accuracy.

A thematic analysis was conducted to extract prominent information and meanings from the qualitative data. Transcripts were analysed in the qualitative analysis program NVivo 12 by a single coder (B.J.).

Our approach employed a reflexive thematic analysis framework as outlined by Braun and Clarke [18]. In reflexive thematic analysis, themes are developed through top-down and bottom-up reasoning. This flexibility was compatible with our study aims. Deductive coding enabled us to organize the data into categories according to our main topics of interest. Inductive coding allowed us to capture novel, unexpected insights from tangential discussions.

First, the transcripts were read multiple times. After becoming familiar with the dataset, the coder created a comprehensive list of thematic codes. These codes were then sorted into thematic categories based on their overarching themes. Following this, themes were extensively reviewed and discussed with a supporting team of researchers at the National Centre for Youth Substance Use Research to ensure they were cognisant and accurately reflected in the dataset. After a final review to reach a consensus, the names and definitions of themes were confirmed. The dataset was then reread to extract frequencies and notable examples.

### 2.5. Ethical Considerations

In this qualitative study, we prioritized the ethical treatment of participants and adherence to established guidelines. Prior to the interviews, informed consent was obtained from all participants, who were assured of their right to withdraw from the study at any point without financial penalty. Participants were also informed they could refuse to answer any questions throughout the interview.

Following interviews, we removed any identifiable pieces of information and deleted video recordings. To provide anonymity, no personal information about participants was recorded. Instead, participants were identified by a randomized number. All data were only accessible to the lead researcher. Participants were provided with a withdrawal form that they could send to the researcher if they wished to withdraw their consent. The participant ID number was attached to the withdrawal form to allow the participant to withdraw their information without compromising their identity.

To minimize risk, the interviewer received extensive training and guidance on how to appropriately and effectively conduct interviews. This included, but was not limited to, creating a non-judgmental atmosphere and identifying signs of discomfort. The questions were developed to be non-accusatory and open-ended. To reduce the risk of misrepresentation, the interviewer clarified they understood the respondents’ meanings instead of relying on their own assumptions.

## 3. Results

We conducted a thematic analysis to identify concurrent discussion points amongst participants in our study. Our analysis produced three themes: (1) factors of engagement, (2) impacts of trading and (3) harm reduction. Differences within themes were represented as subthemes and can be seen in Table 2.

### 3.1. Factors in Engagement

We asked participants about the personal motivations and reasons they decided to trade cryptocurrency. To capture this, we asked participants what prompted their initial and continued decisions to trade. Additionally, we asked about their trading frequency over time and what factors impacted this. Participants described the personal and external factors that influenced their initial and continued decisions to trade. These are represented in Figure 1.

#### 3.1.1. To Make Money

Primarily participants started trading to make money (*n* = 13). Participants sought profit for different reasons. One participant aimed to save for a house deposit, whilst another wanted to buy a car for his mother.

For a few, trading was motivated out of necessity. Experiencing financial difficulties, these participants required cryptocurrency trading as a source of income to make ends meet.


*“Right now, it’s my sole source for finances. My only way of providing. I think that’s the greatest motivation [for trading].”*


Others described aspirations of fortune, motivated to achieve a substantial level of wealth.


*“I will be in some big mansion somewhere with a lot of Lamborghinis, a beautiful family…That’s making it.”*


Participants regularly expressed desires to become successful. For some, it was important not only to succeed in their trading but to be recognized for it as well.


*“It’s all about the large life for me. I just want to be successful like everyone else out there that’s making it big… So that everybody looks up to me.”*


#### 3.1.2. Social Environment

Participants’ social environment greatly influenced their initial beliefs and impressions of cryptocurrency trading. These actors had a considerable influence on participants’ positive expectations of trading. For many, these expectations drove them to cryptocurrency, believing trading would help them achieve their financial dreams.

The majority of participants were exposed to cryptocurrency through friends (*n* = 15). Some participants were recommended trading as a means of making money.


*“A friend of mine who talked about it explained it to me…He told me to do it with expectations that in the next month the prices were going to triple.”*


Many participants described becoming interested in cryptocurrency after seeing their friends’ successes displayed on social media.


*“There’s this friend of mine who had already been acquainted with the cryptocurrency world. I used to see him promote his trades on Instagram. I saw how his life changed overnight. The guy was buying a Mercedes, the latest class. I decided to invest out of fear of missing out.”*


Participants also discussed the social media activity of cryptocurrency influencers (*n* = 7). Influencers are prominent figures in cryptocurrency who often became wealthy through trading. Impressed by their extravagant lifestyle, participants reported developing inflated expectations.


*“The way they show it on the social media. Alive, filled with luxury, buying private jets. This all leaks out into your expectations.”*


Celebrities (*n* = 15) also played a prominent role in participants’ interest in trading cryptocurrency. Due to their success, participants often saw celebrities as authority figures that could guide them to becoming rich. Frequently referenced were Matt Damon and Tom Brady and their appearances in prominent advertising campaigns for cryptocurrency exchanges. The most mentioned figure was Elon Musk, who was regularly applauded for his social influence and material success.


*“I see the rich people like Elon Musk, those people having major decisions in society. Those people are inventing cool stuff, they’re living cool, you know. I think for me, honestly, that’s what I’ve always wanted. And that has always been my motivation to continue trying this stuff.”*


#### 3.1.3. Trading Performance

A factor impacting participants’ continued engagement was their trading performance. Participants who experienced positive returns (*n* = 4) were motivated to continue trading. However, the majority of participants incurred losses (*n* = 11), especially in their first trades. Often entering with high expectations, multiple participants expressed disappointment that outcomes were not as expected. After initial losses, many considered reducing their involvement in trading.


*“But then more losses came. They weren’t adding up and I felt so discouraged. I was almost quitting.”*


Some reported reducing or entirely stopping their trading due to losses. A few participants felt it was not the time due to the state of the market, whilst others stopped because they had no capital to trade.

After stopping due to negative performance, some participants returned with new knowledge or a change in strategy. One trader explained how he returned with greater caution and appreciation for the risks.


*“I did not trade for weeks on end… I just tried to gain more knowledge before I ventured back, while trading with lesser percent of my capital.”*


Multiple participants described an urge to recover their trading losses. Many described continued struggles with urges, with a few characterizing it as addiction.


*“I would say it was like addiction because that urge to try to invest in everything consumed me. I keep having that urge frequently because when I look back at what I’ve lost it kind of t motivated me to keep on trying with the hope that someday I’ll be perfect. Someday I’ll get to recover my losses.”*


#### 3.1.4. Pro-Crypto Attitude

Engagement appeared to be sustained by pro-cryptocurrency attitudes, with many participants describing a passion for cryptocurrency (*n* = 8). These participants viewed cryptocurrency as a new frontier in technology and were committed to remaining involved.


*“I’m motivated because I view myself as a pioneer of an industry that a lot of people don’t understand. I think that starting out something at the ground floor is more fulfilling. It motivates me as I see myself as if I’m in the early stages of the space race. I’m very positive about it. I’m in it for the long term and I will really love if many people will join this space.”*


### 3.2. Impacts of Trading

To examine the impacts of cryptocurrency trading, we asked participants about the positives and negatives that have come from their trading. Whilst we anticipated effects on mental health, our findings revealed a plethora of impacts over a variety of domains.

#### 3.2.1. Positive Impacts

Participants discussed the ways they were positive influenced by their trading. Positive impacts typically occurred between 2020–2021. During this period, some participants (*n* = 7) saw improvements in their financial situation.


*“The positives, maybe the profits. Sometimes you sell when it’s high, and you have some bitcoin spare. It’s a secure way or a better way of keeping my money working, making money, instead of just keeping the money in the bank.”*


Often this was associated with positive impacts on their mood (*n* = 5), with participants reporting feelings of elation, happiness and excitement when their trades performed well. These profits also helped reduce mental distress by alleviating financial difficulties. Others (*n* = 4) reported positive impacts on their relationships. One participant felt these trading profits helped strengthened their relationship.


*“I even told you that I bought my lady a vehicle. It really cemented our relationship because at that time I was not in a position to marry her. But it really helped us move forward because anything she wanted, she could get. You get some extra love when you provide as a man.”*


#### 3.2.2. Negative Impacts

However, participants also experienced problems due to their trading. The extent of harm appeared related to their exposure to the market, with highly invested individuals incurring greater harm. These were largely related to the recent downturn in the market, with the most negative impacts occurring in 2022. Participants experienced negative impacts to their mental health (*n* = 14), financial situation (*n* = 10), relationships (*n* = 10), occupational capacity (*n* = 4) and sleep (*n* = 6).

Participants frequently described feeling drops in their mood, experiencing sadness and regret after losing money. For some, this resulted in missing out on work or school.


*“When I experienced this I wasn’t able to go to classes for some days…I think because I was feeling bad because of the money I lost. I just decided to stay at home. I just wanted to be alone.”*


Additionally, a few participants described feelings of irritability when their investments performed poorly. This occasionally cumulated in outbursts that were usually directed at people close to them.


*“There’s also anger management issues. You can just pass out of nowhere without any provocation you have. It is noticeable. Anyone can say that. This guy’s not okay.”*


Participants also observed these effects amongst their trading friends.


*“[My friends] moods out of nowhere, they just become not so pleasing people. Like a person may just start ranting. Even if we meet over some drinks, like one of them just broke down. That wasn’t the best of days, but I understand. So I might say it has affected their mood, their moods very, very much.”*


The impact on mood was most dramatic following sudden, significant financial loss.


*“There’s this Netflix documentary that people invested in some company and the guy just went away with the all the funds and was living a rich life. Such cases have been reported worldwide and there was a time that I had stakes in one of those companies…All that made me depressed. It’s like a heartbreak. You stay in bed the whole day and nothing else mattered.”*


A few described being unable to support their families due to their trading losses. For some, this had a significant strain on their relationships.


*“Her condition was so critical that my family started complaining. How many years I have worked and I’m not able to provide that particular amount of money to help my mom…When my family realised I was into crypto my mom especially was very angry.”*


To fund their trading, some participants borrowed money from friends and family. A couple of participants lost these borrowed funds trading, leaving them in debt. This placed enormous strain on their mental state and relationships.


*“There’s a time where I lost a huge amount of money. It was money that I had loaned from a friend, so I remember that time I almost committed suicide. I almost committed suicide. It’s something which is sad to say.”*


Participants also experienced problems due to the time they spent on trading activities. Some reported feeling preoccupied with trading even when away from the market. Additionally, the time participants spent analyzing, researching and conducting trades impacted other areas of their life.

Commonly participants described reductions in sleep due to staying up late to monitor the markets. A couple of participants reported using substances to stay awake.


*“Sometimes I don’t sleep at all and that really affected me. There was a period when I had to use some drugs, you know, to keep me awake, just to study the market. I had to visit a medical practitioner, just to get back into my former self.”*


A few participants referenced the 24/7 nature of the market as the reason for their changes in sleep.


*“The markets doesn’t close like stocks. You always have to keep an eye out. It keeps you guessing and it keeps you calculating…so there’s an issue with insomnia.”*


For some, this had a significant impact on their non-cryptocurrency relationships. Participants reported having less time to hang out with friends and family members. This would sometimes result in complaints from loved ones. For one trader it threatened to deteriorate his marital relationship.


*“My wife even threatened at some point, I think it was during COVID, she threatened to go. To leave me because once again, I get on the phone with my friends and all we can think about is maybe let’s trade this, or let’s sell this, let’s buy that.”*


### 3.3. Harm Reduction

We were interested in the ways cryptocurrency traders coped with distress. To capture this, we asked participants how they dealt with distress from the market and their recommendations to others. Additionally, participants also discussed ways to reduce harm in general. This was also included in the theme.

Following economic loss, participants (*n* = 13) found it beneficial to take a break from the markets and focus on life outside trading. This was helpful in combatting emotional distress after incurring losses. During this time off, participants typically engaged in other hobbies and activities to keep themselves occupied. These included exercising, video gaming and spending time with friends. Participants also engaged in activities that helped them relax and distress, such as listening to music or taking a walk.


*“Sometimes when things aren’t that good, I just completely stop everything. I get out if it’s during the day. After a while when I’m a bit relaxed, then I may resume.”*


Social support appeared to play an important role in alleviating mental distress. Several participants mentioned the support and encouragement they received from people close to them and their role in coping with trading losses (*n* = 7). Some found solace in conversation with their significant other. Others benefitted from talking with other traders.


*“I have these friends who overall have gone through the same experience. I think that made me to become stronger. I felt like, I’m not going through this alone. I think that made me overcome all these suicidal thoughts.”*


Several participants gave suggestions on how to reduce the risk of financial loss when trading cryptocurrency (*n* = 10). Participants recommended investing only what you can afford to lose.


*“Don’t risk what you can’t afford to lose. Position yourself to the point that if you lose this money, or if you blow this account, you won’t cry. You won’t miss milk in the house or miss your bills.”*


To reduce risk, participants commonly shared recommendations for responsible trading. These included not overtrading, not trading on impulse and conducting research and analysis before making decisions.


*“I would say the best trader is the one who goes with caution, where you doesn’t just rush or trade impulsively. Because trading like that causes a lot of problems such as negatives in terms of your health and also in terms of your economic potential.”*


Multiple participants discussed the importance of increasing awareness of the risks of trading (*n* = 9). Some felt they were not adequately informed of the risks when they started trading. Participants placed responsibility on those guiding newcomers to give them an honest representation of their experiences.


*“I think people who are guiding others as beginners, should tell them about every aspect of crypto, the bad, the risks they have to take and the potential loss. Because sometimes people just tell others about the times they gain without telling people about times they lose.”*


Participants also placed responsibility on advertisers and public figures in cryptocurrency. Many participants felt cryptocurrency and its risks were misrepresented in advertising and endorsement.


*“The possible gains and risks are misrepresented. I’ve seen celebrity adverts about cryptocurrency. I’ll be like: “No! This shouldn’t be.” This is a trap, for beginners. Having your favourite celebrity shill coins out. There should be a way for the possible gain and loss to be represented properly and let people decide for themselves whether they want to venture in.”*


As a result, some participants cautioned against following the advice of influencers, warning of their intentions.


*“Think social influences who advertise and try to convince people go into cryptocurrency and get a particular coin. They have been paid to convince the public. So they try to do everything possible, to make sure that you get into crypto. And by doing this, they give you the advantages, the positivity and they fail to tell you about the downsides.”*


## 4. Discussion

Through qualitative interviews, we examined the experiences of individuals who have incurred harm due to cryptocurrency trading. One of our aims was to investigate how participants were affected by their trading.

Our study revealed the significant impact that cryptocurrency trading has across multiple domains of their lives. They extended beyond their financial situation, impacting their mood, relationships and physical and mental health.

Whether these impacts were positive or negative depended largely on the performance of their trading. Positive experiences were typically enjoyed during 2020–2021, a period where cryptocurrency increased by over ten times in market cap [1]. During this time, participants saw improvements in their finances, mood and personal relationships. Conversely, when participants lost money, their moods changed dramatically, their stress increased, and their relationships suffered. The mental health of those who suffered dramatic losses was alarming, with multiple individuals reporting feelings of depression and one experiencing suicidality.

Considering the volatility of the cryptocurrency market, it is concerning that traders’ quality of life is so closely tied to the market’s performance, as their lives can fall apart very quickly. This indicates the need for further research on the impact of trading on individual investors, particularly following significant market downturns. Our findings also emphasize the important of developing support services for individuals negatively impacted by their trading.

Our study significantly contributes to our overall understanding of the impact of financial trading on individual investors. Whilst there has been investigation conducted in other financial markets, such as the stock market, these have typically focused on effects at a population level [19,20]. Additionally, these studies are typically conducted after significant stock market crashes in large financial markets, which inflict reductions in the quality of life for people overall [21]. These findings should encourage investigation into the experience of traders in other financial markets, which have experienced downturns of similar magnitude, such as the dot.com crash in the early 2000s which saw many tech companies go bankrupt [22]. 

We also aimed to explore how individuals coped with the impacts of cryptocurrency trading. Consistent with existing research [23], we found that social support was beneficial in coping with emotional distress. Participants also benefitted from taking a break from the markets, often engaging with other hobbies and activities. As these involve diverting attention away from the problem rather than directly dealing with them, these might be considered avoidant coping mechanisms [24]. Whilst these are often considered maladaptive [25,26], avoidant coping mechanisms can be valuable as they allow for the gradual dosing of emotional impact over time [27]. This may protect the mental well-being of traders after dramatic financial loss and prevent them from making rash decisions. However, the long-term effectiveness of avoidance-based strategies is heavily context-dependent [28], and thus requires further research.

Additionally, we captured factors relevant to engagement with cryptocurrency trading. Our findings found desire and expectations to make money as primary motivators for trading cryptocurrency. This is consistent with previous findings in traditional financial markets, with profit expectation being a primary driving factor of investment [29]. A difference, however, is that expectations of returns in stock market investment are commonly based on financial attributes of the investment, such as dividends, yields and management [30,31]. We found, however, that profit expectation for cryptocurrency traders was primarily based on information from their social environment.

Cryptocurrency promotions played a substantive role in forming these initial expectations. With advertisements promoting trading as a way of achieving a glamorous lifestyle, these messages were especially influential on individuals who aspired to become rich. They also appeared effective on those with limited investment experience, who are likely to be vulnerable to economic loss due to their lack of knowledge [32].

Many participants felt that the promotions for cryptocurrency trading were dishonest and did not accurately convey the risks involved. Given the potential dangers of investing in cryptocurrency, it is important that the risks are clearly and accurately represented. Future studies should explore how cryptocurrency trading is depicted in these promotions and their influence on people’s investment decisions. It also emphasizes the importance of increasing awareness of the risks at a community level.

Our findings justify concerns about the potential for cryptocurrency trading to be addictive [33]. After facing initial loses, some participants continued trading due to urges to make back the losses. This is commonly seen in problem gamblers [16], indicating similarities between the behaviours. Participants shared other behaviours with problem gamblers, many of which feature as diagnostic criteria [34]. These include preoccupation with trading, borrowing money and damage to relationships.

Our study, however, revealed some differences between behaviours. Unlike gambling, trading was not used as a distraction or way to cope with outside life events [35]. Additionally, individuals with problem gambling often develop tolerance and thus have to gamble increasingly large amounts to achieve the same level of excitement [36]. However, our study did not find this for trading. Trading, in general, did not appear to be motivated by intrinsic reasons. Participants did not describe trading due to boredom, stress or loneliness [37], which is also common amongst individuals who gamble. Additionally, after initial losses, many participants reduced or stopped trading altogether. This separates it from other problematic behaviours, where individuals continue to engage in the behaviour despite negative consequences. Further research is required to investigate the characteristics of cryptocurrency trading as an addictive behaviour and assess its relationship to problem gambling.

### Limitations

There were some limitations to our study’s design. Firstly, we conducted interviews following a significant contraction in the market. Therefore, our findings may not reflect the experiences of cryptocurrency traders during upswings in the market cycle. Furthermore, our sample contained only individuals residing in the United States and Australia, meaning findings cannot be generalized to cryptocurrency traders worldwide.

Aiming to capture the adverse impacts of cryptocurrency trading, we specifically recruited those who self-reported problems due to trading. However, this meant our findings might not reflect the experiences of cryptocurrency traders overall. Additionally, our findings do not offer insight into the prevalence of which harm occurred.

## 5. Conclusions

Because large fluctuations in the cryptocurrency market are common, it is important to investigate the experience of individuals impacted. To achieve this, we interviewed cryptocurrency traders who had experienced problems due to their trading following a downturn in the market.

We found that participants’ experiences were related to their trading performance. This is concerning, given the potential for economic loss. The recent reduction in the market saw negative impacts on many participants’ financial situation, relationships, mental health and occupational capacity. This emphasizes the importance of administering intervention strategies following downturns in the cryptocurrency market. It should also encourage further research into the impact of trading in other financial markets.

Our findings indicate the protective role social support plays against emotional distress. Additionally, they revealed the utility of breaks from the market in dealing with trading losses. Our study also revealed the significant impact cryptocurrency promotions have on individuals’ expectations and intentions toward cryptocurrency trading. With many participants feeling misled, it is important to investigate the content of these promotions and their depiction of the risks. Increased pressure should be placed on advertisers and promoters of cryptocurrency trading products to accurately represent the risks of trading. Additionally, treatment services should look to integrate programs and facilities to assist affected individuals, especially following significant downturns in the market.

In addition, with some participants describing struggles with addiction, investigation should be conducted into the prevalence and nature of cryptocurrency trading as a problematic behaviour.

## Figures and Tables

**Figure 1 ijerph-20-05833-f001:**
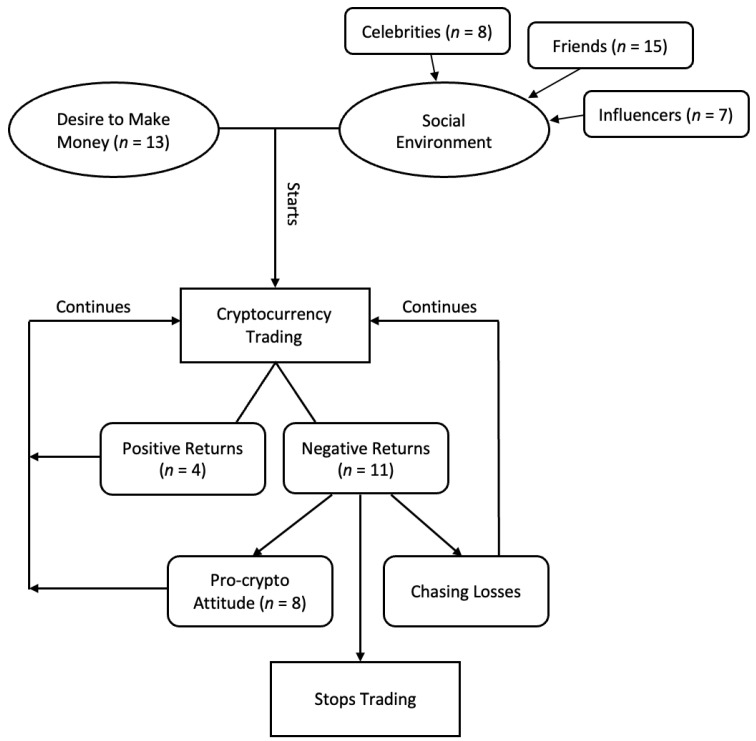
Factors in engagement of cryptocurrency trading.

**Table 1 ijerph-20-05833-t001:** Demographic characteristics of participants.

Characteristic	*n*	%
Gender		
Male	13	76.5
Female	4	23.5
Age in years (M = 25.6, SD = 4.16)		
18–22	5	29.4
23–27	5	29.4
28–32	7	41.2
33 +	0	0
Highest Educational Level		
High-School Diploma	2	11.8
Associate Degree	2	11.8
Bachelor’s degree	11	64.7
Post-Graduate Degree	2	11.8
Country of Residence		
US	12	70.6
Australia	5	29.4
Ethnicity		
White	3	17.7
Hispanic or Latino	1	5.9
Asian	2	11.8
Black or African American	8	47.1
Prefer not to state	3	17.7

**Table 2 ijerph-20-05833-t002:** Themes and subthemes of thematic analysis with codes and frequencies.

Theme	Description	Subtheme	Recurring Codes
Factors in Engagement	Discussion on contextual and personal factors that motivated and sustained cryptocurrency trading behaviours.	Desires to Make Money (*n* = 13)	-Trading to make money -Motivated by dream of been rich -Making money to support family and friends -Wanting to be financially independent -To alleviate financial struggles
		Social Environment (*n* = 15)	-Directly recommended to trade by friends -Interested in trading due to friends’ success -Attracted to trade from celebrity endorsement -Celebrities as aspirational figures -Encouraged by lifestyle portrayed by influencers
		Trading Performance (*n* = 14)	-Continued trading due to profits -Cutting down or reducing trading due to losses -Chasing losses
		Pro-crypto attitude (*n* = 8)	-‘Crypto is the future’ -New frontier in financial technology -Desire to be a pioneer of the crypto industry
Impacts of Trading	Discussion on the positive and negative impacts that have resulted from participants cryptocurrency impacts.	Positive	
		Financial (*n* = 7)	-Improve financial situation -Supporting self through crypto trading
		Mood (*n* = 5)	-Elation, excitement and joy after profits -Alleviation of stress due to financial struggles
		Relational (*n* = 5)	-Made trading friends -Trading strengthened friendship groups -Profits solidified relationship with significant other
		Negative	
		Financial (*n* = 10)	-Worsened financial situation -Put trader in debt -Unable to support family
		Relational (*n* = 10)	-Damaged relationships due to financial loss -Complaints from others about time spent on market -Conflict with family/friends that traders loaned from
		Mental Health and Wellbeing (*n* = 14)	-Losses negatively impact affect -Depression after significant financial loss -Irritation resulting in occasional outbursts -Distress from poor trading performance -Anxiety from market volatility
		Occupational (*n* = 4)	-Absent from work/school due to low mood -Hard to balance trading and school
		Sleep (*n* = 6)	-Staying up late to monitor markets -Reduced sleep due to time researching -Difficulties sleeping due to mental distress
Harm Reduction	Discussion of methods to reduce mental distress and risk of financial loss while participating in the cryptocurrency market.	Life Outside Trading (*n* = 13)	-Take a break from markets -Listen to music and take a walk -Engaging in hobbies and activities -Focus on work
		Social Support (*n* = 7)	-Comfort from loved ones -Support from trading friends -Discussing stress with therapist
		Responsible Investment (*n* = 10)	-Invest what you can afford to lose -Conduct thorough research and analysis before trading -Trade with caution and with a plan -Don’t overtrade -Ask other traders for advice
		Increase Risk Awareness (*n* = 9)	-Crypto misrepresented by adverts -More awareness required of the risks -Friends and adverts responsible for educating on risks -Be way of influencers intentions

## Data Availability

Data sharing is not applicable to this article.

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
