# Peer review of "“Buy High, Sell Low”: A Qualitative Study of Cryptocurrency Traders Who Experience Harm"

_ijerph, 2023, doi:10.3390/ijerph20105833_

Round 1

Reviewer 1 Report

Some treat cryptocurrencies as an investment, others as a game and hobby, but some of us as a gambling. Addiction specialists are increasingly pointing out that the highly speculative nature of digital currencies, along with the possibility of “winning” or “losing” significant assets on investments, bears the hallmarks of addiction. Furthermore, it should be empasized that they have only been in the mainstream for a very short time. Therefore, it makes sense that the dangers are not explicitly covered, as in the case of gambling addiction. In the context of the above, it seems likely that with the development of cryptocurrencies - which has so far proceeded at a pace similar to the internet in its early days - addictions will also grow with them. It should also be noted that cryptocurrency trading takes place 24/7. Unlike conventional financial markets, the cryptocurrency market never closes. Just like in sports betting, where you can only bet when the event is on, there are no “free hours” in cryptocurrencies. The most addictive is the high volatility in the cryptocurrency market. For example, buying a cryptocurrency on a hill can cause the effect of the so-called. "high". Addiction to cryptocurrency trading can therefore be as negative as gambling or drug addiction.

In the light of this information, the presented manuscript concerns a very important issue and, additionally, can bring us much closer to a not fully understood social problem. The manuscript was properly structured in accordance with the requirements of the journal.

The strongest point of the manuscript is the precise and very detailed description of the selected research methods and the results obtained. Any additional information in the form of questionnaires or tables significantly increases the value of the presented manuscript. In my opinion, the Authors should:

Shorten the introduction, including only the most necessary information

In addition, the number of people surveyed in the research seems to me insufficient (not a very representative group). Add a comment to this point.

Most of the Authors' observations are very accurate, but they are already confirmed by other studies. In my opinion, the Authors should clearly emphasize how their research differs from others and what is new in them.

Reviewer 2 Report

Thank you very much for the opportunity to review the manuscript. I congratulate the authors on their approach to the subject. The article has an excellent structure and transparency, an excellent analytical part, and an interesting discussion. 

I suggest tidying up the methodological part and clarifying a few ambiguities. Here are some issues that will help the article's authors prepare the final version of the manuscript. 

1 Introduction

Line 84. In this paragraph, it seems reasonable to me to add a sentence referring to whether cryptocurrency trading can lead to non-substance addiction, like gambling. I understand this is not a topic of consideration for the manuscript's authors, but it would be good to know that the authors are aware of this phenomenon. 

Line 118-122. First of all, I miss an unequivocal statement of the purpose of the study. Perhaps the word "primarily" would be worth replacing with "main". The current arrangement of the study's objectives can be read like a child's rant: I want this. And this. And this. And also, this. And this.

In addition, the authors say they want to study "effects across other domains in life" - which ones? In this paragraph, shouldn't the past tense be used? Have we explored rather than want to explore? 

2 Materials and Methods

This section needs to build a new structure and organize and state the content, such as Study Design, Questionnaire of the Interview, Data Collection, Subject Recruitment, Sample Characteristics, Data Analysis, and Ethical Considerations.

Line 129. The phrase "experienced problems" is ambiguous - it could be mental problems related to this and the technical aspects of cryptocurrency trading, and I think it is worth specifying "problems". 

Line 140. It would be appropriate to separate the section on the Interview Questionnaire. Describe what the semi-structured nature of the interview consisted of. From the attached tool, the interview was highly structured, and as a methodologist, I can't answer what the semi-structured consisted of, and I would love to learn from the article. 

Lines 151-152. Shouldn't this sentence be in Data Analysis?

Lines 152-153. This sentence should be separated as Ethical considerations. It is worth re-emphasizing here that the interview participants gave informed consent to participate. 

Line 157. This sentence should be in the Methodology section.

Reviewer 3 Report

This paper reports the findings of a small Australian qualitative study into the potential harm associated with cryptocurrency trading. A total of 17 people who had experienced some problems with crypto were asked to respond to a semi-structured interview. Findings were generally as would be expected. Excessive engagement was associated with financial losses, stress and anxiety, disrupted sleep, strain on personal relationships and competition with other commitments.

I think that this is an important topic which will become more relevant over time. Many of the observations seem quite valid. The first part of the paper showed a good understanding of the emergence of cryptocurrency. The method is Ok and the Results are generally analysed and presented in a way appropriate for a qualitative paper.

However, there are a few aspects of the paper which need to be addressed to make it more rigorous.

The first thing I noticed was that it’s quite under-referenced. There have been a number of papers on the psychology of cryptocurrency trading and these are not even mentioned. I am aware of other Australian papers:

Delfabbro, P., King, D. L., & Williams, J. (2021). The psychology of cryptocurrency trading: Risk and protective factors.. Journal of behavioral addictions, 10(2), 201-207.

Delfabbro, P., King, D., Williams, J., & Georgiou, N. (2021). Cryptocurrency trading, gambling and problem gambling. Addictive Behaviors, 122, 1-6.

There is work in Canada which provides important conceptual parallels between gambling and speculative trading

Arthur, J., Williams, R., & Delfabbro, P. (2016). The conceptual and empirical relationship between gambling, investing, and speculation. Journal of Behavioral Addictions, 5(4), 580-591.

I have also reviewed several important German papers which are probably out by now.

The second issue, and this really follows from the lack of depth in the literature review, is that the paper does not really engage with the psychology of cryptocurrency. Why is it tempting and potentially addictive? How is it different from conventional speculative trading or tech stocks which  also dropped 60-70% in 2022.  The drop in Amazon was one of the largest capital write-downs in history? I would expect to see more discussion of FOMO effect, the nature of mobile and tablet technology and how it makes trading so fast and convenient; more on Youtube and social media influencers (mentioned, but this was a major feature of the meme mania in 2021). The authors also do not distinguish between trading and investing. Trading is much riskier than investing across the 4 year cycle. The paper provides a somewhat unbalanced account of the crypto market. People have clearly done well- need to explain how- and then how this compares with short-term speculative trading.

Minor points

Section 1.1 The total market cap went over 3 trillion. Just check the Coinbase statement: are you sure it’s 7.4T. I looks like 7.8 billion. If so, that’s a big difference.

1.2 Note that many tech stocks (including Meta) were as bad some of crypto projects and that we saw similar drops in the dot.com crash in the early 2000s. It’s not all a crypto phenomenon.

p. 12  Chasing losses- it’s similar in crypto, but it’s also a good investment strategy to drop cost-bases for quality projects in bear markets?

p. 12, 13  Never use ‘space’ in academic papers. It’s managerial jargon.

Otherwise, an interesting paper.

Also avoid ‘This is concerning….” (conclusion). Not really the role of academic research to be making emotive statements to the reader about what upsets them. Just need to state the facts. Concerning to the government?

Interventions almost need to be brought in during the highest risk periods which involve higher prices (the bull runs?). You’ll see the harm in the bear markets, but the damage will be done when everyone is looking at green candles?

The final bit is a little vague. Who would be providing the consumer information? What type of things? In what media or in what context? Any implications for treatment services?

Round 2

Reviewer 3 Report

The authors have satisfactorily addressed the points I raised in my original review.